# Soft Constraints, Strong Solutions: Optimizing Intra-Operator Parallelism for Distributed Deep Learning

## Abstract

As deep learning models grow in size and complexity, efficiently mapping their computations onto distributed hardware is a central challenge for systems and compiler design. A key technique for addressing this challenge is intra-operator parallelism, which partitions individual operations across multiple devices. To accelerate research on automated intra-operator parallelism, Google curated a benchmark suite of 25 large-scale instances drawn from real production workloads including Graph Network Simulators, U-Nets, diffusion models, and Gemma 1 and Gemma 2 language models, and organized the ASPLOS/EuroSys 2025 Contest on Intra-Operator Parallelism for Distributed Deep Learning. The contest formalized intra-operator parallelism as a constrained combinatorial optimization problem in which each computational-graph node must be assigned an execution strategy that minimizes compute and communication cost while satisfying strict time-varying memory limits. This paper presents the winning solution. We show that relaxing the hard memory constraints enables the problem to be reformulated as a Cost Function Network optimization task. Building on this idea, we develop a solver that combines adaptive penalty-based relaxation with efficient Cost Function Network optimization. The method quickly produces feasible strategies with costs near the global optimum on nearly all benchmark instances, consistently outperforming XLA, the production-grade compiler used in TensorFlow and JAX, often by orders of magnitude.

## 1 Introduction

The scale and complexity of modern deep learning models have made distributed execution a necessity. As models grow to encompass billions of parameters, they demand enormous compute and memory bandwidth, often surpassing the capacity of any single device. Efficiently partitioning and scheduling these computations across device meshes has therefore become a critical challenge for both system designers and compiler developers.

A key component for addressing this challenge is *intra-operator parallelism*, which slices individual tensor operations (e.g., matrix multiplications, elementwise ops) across multiple devices. This strategy enables fine-grained parallelism and better hardware utilization, but it also introduces considerable communication costs due to split and merge operations (Zheng et al., 2022). In state of the art schemes like Alpa (Zheng et al., 2022), it is combined with *inter-operator parallelism*, which partitions the computation graph into larger sequential stages that are mapped across devices, typically in a pipelined fashion.

Alpa follows a two step approach where first inter-operator parallelism is optimized, and then intra-operator parallelism is configured for each stage independently. This requires repeatedly solving many intra-operator parallelism problems, each corresponding to a subgraph of the original computation graph. Thus, efficiently optimizing intra-operator parallelism is crucial for for the overall compiler performance.

However, existing solvers failed to optimize large models on large accelerator grids within practical time limits, hindering the adoption of automatic intra-operator parallelism in production systems. To address this gap, Google curated a benchmark suite of 25 instances derived from real-world models

and organized the ASPLOS / EuroSys 2025 Contest on Intra-Operator Parallelism for Distributed Deep Learning (Moffitt & Fegade, 2025b). The contest formalized intra-operator parallelism as a constrained combinatorial optimization problem, where each node in a computational graph must be assigned an execution strategy that minimizes total cost (compute + communication) while ensuring that time-varying memory usage remains within a global limit. As described in the contest report, these memory constraints significantly increase the problem's complexity (Moffitt & Fegade, 2025b). In fact, without these constraints the problem naturally lends itself to *Cost Function Networks (CFNs)* optimization (Allouche et al., 2010; Rossi et al., 2006).

CFNs, also known as weighted Constraint Satisfaction Problems, are a mathematical model derived from classical constraint satisfaction problems by replacing hard constraints with cost functions (Allouche et al., 2010). Each cost function assigns a non-negative integer cost to every possible combination of values over a subset of variables. CFNs naturally capture the graph structure of the optimization objective but struggle to enforce the strict global memory constraints. To overcome this limitation, we use a Lagrangian relaxation inspired approach that integrates the memory constraints directly into the cost model. This relaxation enables our algorithm to scale to graphs with tens of thousands of nodes. We obtain tight lower bounds for most instances, confirming that our solutions are close to the global optimum. Our solver consistently produces more cost-efficient solutions than XLA (XLA Developers, 2025), the production-grade compiler employed in TensorFlow (Abadi et al., 2016) and JAX (Bradbury et al., 2018), often by orders of magnitude, as shown in the evaluation section.

## 2 RELATED WORK

Efficient parallelization of machine learning workloads has been the focus of extensive research. Early work in distributed training combined data, model, and domain parallelism to scale networks across devices (Gholami et al., 2018; Wang et al., 2019; Rajbhandari et al., 2020). While these approaches exposed multiple axes of parallelism, they often relied on manual configuration or static partitioning strategies.

To automate parallelism decisions, compiler-based systems were developed. GShard (Lepikhin et al., 2021) and GSPMD (Xu et al., 2021) introduced planner-driven compiler transformations for device sharding. Piper (Tarnawski et al., 2021) unified placement and graph partitioning within a shared framework. More recent systems such as Alpa (Zheng et al., 2022), nnScaler (Lin et al., 2024), and Liger (Du et al., 2024) use cost models and solver-based optimization to automatically configure both inter-operator and intra-operator parallelism. Notably, Alpa's core functionality has been integrated into XLA (XLA Developers, 2025), adding support for automatic sharding and distributed training (Alpa Developers, 2023).

In parallel, research on sharding tensor and optimizer states has reduced memory and communication costs during training (Xu et al., 2020; Jiang et al., 2023; Shi et al., 2023; Zhao et al., 2023), helping improve training scalability. These techniques are complementary to, but distinct from, the strategy assignment problem we focus on.

This paper addresses the joint optimization of cost and memory usage within intra-operator parallelism. We propose a usage-constrained relaxation approach that integrates memory constraints directly into the cost model. Our method is based on Lagrangian relaxation (Yurkiewicz, 1985; Laue et al., 2020; 2022), which absorb constraints into the objective using penalty terms, but avoids some of its limitations. Unlike standard Lagrangian relaxation techniques, this algorithm does not require lower bounds and is specifically designed to reliably find feasible solutions.

## 3 INTRA-OPERATOR PARALLELISM AS DISCRETE OPTIMIZATION PROBLEM

The execution of deep learning models can be represented as a directed acyclic graph (DAG), where each node corresponds to an operator (e.g., convolution, matrix multiplication or softmax) and edges represent data dependencies between operators (e.g. the second layer depends on the output of the first layer). In *intra-operator parallelism* one or more tensors are partitioned across multiple devices to accelerate the execution of individual operators. During its optimization a compiler typically generates a set of candidate partitioning strategies for each operator and estimates their computation

cost, communication cost, and memory usage profile (Moffitt & Fegade, 2025b). Afterwards, an optimizer must select one strategy per operator to minimize the total cost while satisfying global memory constraints. In our work we focus on this downstream *strategy selection* task, which was formalized for the contest as given in the following (Moffitt & Fegade, 2025b). However, any solver for the formal problem can be plugged into a full compiler pipeline that also includes strategy generation, as is already done in XLA.

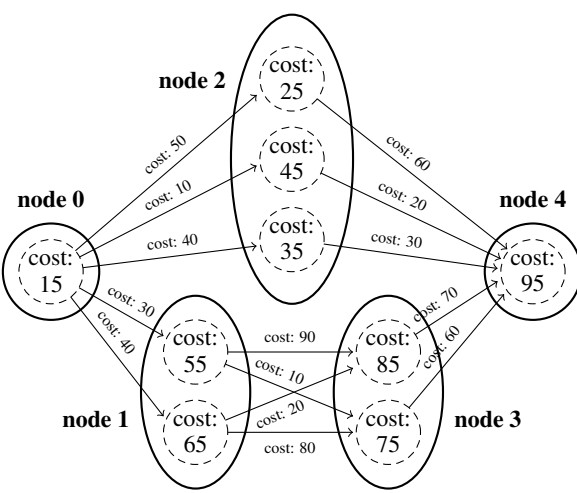

Figure 1: Problem graph with annotated node and edge costs.

Let $G = (V, E)$ be a directed acyclic graph representing a deep learning computation, where each node $v \in V$ corresponds to an operator and each edge $(u, v) \in E$ indicates a data dependency from operator $u$ to operator $v$. For example in Figure 1 node 0 could be a reshape operation, whose result is passed to two independent convolution nodes 1 and 2. Afterwards, node 3 applies another convolution onto the result of node 1. Finally, the independent convolutions get aggregated with a max or sum operator in node 4.

The overall computation is divided into $T$ discrete time steps, each operator node $v$ is active during a specific time interval $[b_v, e_v] \subseteq [0, T]$. Moreover, for each node $v$ we are given a set of $k_v$ candidate partitioning strategies $S_v = \{1, 2, \ldots, k_v\}$. The objective is to find a strategy assignment $s = (s_v)_{v \in V}$ that minimizes the total cost:

$$C(s) = \sum_{v \in V} c_v(s_v) + \sum_{(u,v) \in E} c_{u,v}(s_u, s_v). \qquad (1)$$

subject to the memory constraint that, at any time $t$, the total memory usage of all active nodes does not exceed a global limit $M$:

$$\sum_{v \in V : b_v \leq t \leq e_v} m_v(s_v) \leq M, \quad \forall t \in \{0, 1, \ldots, T\}. \qquad (2)$$

Here each node $v$ has an associated

$$\text{computational cost } c_v : S_v \to \mathbb{N}_{\geq 0},$$
$$\text{and memory usage } m_v : S_v \to \mathbb{N}_{\geq 0}$$

and each edge $(u, v) \in E$ has a communication cost

$$c_{u,v} : S_u \times S_v \to \mathbb{R}_{\geq 0} \quad \text{for each } (u, v) \in E.$$

Figure 1 illustrates an example problem with annotated node and edge costs for different strategies. Choosing the assignment $(1, 1, 3, 2, 1) \in S_0 \times S_1 \times S_2 \times S_3 \times S_4$ results in a total cost of 445. Figure 2 visualizes the memory usage profile over time for this particular solution.

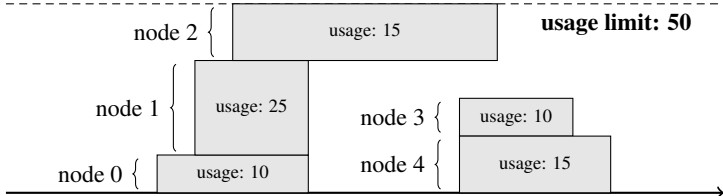

Figure 2: Memory usage over time for the assignment $(1, 1, 3, 2, 1)$. The memory limit is never exceeded, indicating feasibility.

# 4 PROPOSED METHOD

The memory constraints significantly increase the problem's complexity (Moffitt & Fegade, 2025b), making it challenging to solve large instances within practical time limits. Therefore, it is natural to consider the Lagrangian relaxation of the problem, which incorporates the memory constraints into the objective function using non-negative multipliers $\lambda \in \mathbb{R}^T$:

$$\mathcal{L}(\lambda) = \min_s C(s) + \sum_{t=0}^{T} \lambda_t \left( \sum_{v \in V : b_v \leq t \leq e_v} m_v(s_v) - M \right) \tag{3}$$

for $\lambda = (\lambda_1, \ldots, \lambda_T) \in \mathbb{R}^T$ and $\lambda_i \geq 0$.

Because $m_v(s_v)$ does not depend on $t$, we can rewrite the penalty term as:

$$\sum_{t=0}^{T} \lambda_t \left( \sum_{v \in V : b_v \leq t \leq e_v} m_v(s_v) - M \right) = \sum_{v \in V} m_v(s_v) \sum_{t=b_v}^{e_v} \lambda_t - M \sum_{t=0}^{T} \lambda_t. \tag{4}$$

Which we can shorten further by defining $\lambda_v = \sum_{t=b_v}^{e_v} \lambda_t$ and $M_\lambda = M \sum_{t=0}^{T} \lambda_t$. Thus, the Lagrangian relaxation becomes:

$$\mathcal{L}(\lambda) = \min_s \sum_{v \in V} \left( c_v(s_v) + \lambda_v m_v(s_v) \right) + \sum_{(u,v) \in E} c_{u,v}(s_u, s_v) - M_\lambda \tag{5}$$

for $\lambda = (\lambda_1, \ldots, \lambda_T) \in \mathbb{R}^T$ and $\lambda_i \geq 0$.

This relaxation has the form of a Cost Function Network (CFN)(Allouche et al., 2010; Rossi et al., 2006). And allows us to leverage highly efficient CFN solver, that exploit the underlying graph structure of the problem and are well used to binary cost functions as given by the edge costs $c_{u,v}(s_u, s_v)$. A CFN consists of discrete variables $\{s_1, s_2, \ldots, s_n\}$ with finite domains $S_1, S_2, \ldots, S_n$ and a set of cost functions $\mathcal{F}$, where each $f \in \mathcal{F}$ maps assignments on a small subset of variables to a non-negative cost. The goal is to find an assignment $s$ minimizing the sum of all costs

$$C(s) = \sum_{f \in \mathcal{F}} f\left( s_{|f} \right). \tag{6}$$

In $\mathcal{L}(\lambda)$, each node $v$ induces a unary term $f_v(s_v) = c_v(s_v) + \lambda_v m_v(s_v)$ and each edge $(u, v) \in E$ induces a binary term $f_{uv}(s_u, s_v) = c_{u,v}(s_u, s_v)$, while the constant $-M_\lambda$ can be ignored for optimization. The relaxed objective is therefore a CFN with unary and pairwise costs.

By solving the dual problem $\max_\lambda \mathcal{L}(\lambda)$, one can obtain lower bounds on the optimal solution of the original problem. Unfortunately, even the Lagrangian relaxation remains hard (Shimony, 1994) and optimizing the dual requires good lower bounds, which are harder to obtain, than a good solution. Additionally, solutions found in the relaxed problem may violate the original memory constraints, necessitating further adjustments to ensure feasibility. Even when the Lagrangian dual is maximized, there is no guarantee that the solution satisfies the original memory constraints. Moreover, the common approach for maximizing the dual increases $\lambda_i$ for all time steps $i$ where the memory constraint is violated and decrease it otherwise. In practice this often leads to decreasing a $\lambda_v$ for a

node that violates a memory constraint violation, because the non violated constraints outnumber the violated ones. Since we are interested in finding feasible solutions rather than lower bounds, we propose a different approach based on per node usage penalties $w_v$ instead of per time step multipliers $\lambda_t$.

$$\arg\min_s \sum_{v \in V} \left( c_v(s_v) + w_v m_v(s_v) \right) + \sum_{(u,v) \in E} c_{u,v}(s_u, s_v) \quad \text{for } w_v \geq 0. \tag{7}$$

To find effective values of $w_v$, we employ an adaptive algorithm that iteratively adjusts the weights based on feasibility feedback from the solver (see Algorithm 1). This algorithm gradually aligns the

---

**Algorithm 1:** Adaptive Weight Optimization

---

**Input:** Initial strategy cost functions $c_i(x_i)$, usage profiles $u_i(x_i)$, global usage limit
**Output:** Feasible, low-cost strategy assignment $A$
1 Initialize $w_v$ such that solving the relaxed CFN yields a feasible assignment $A$
2 **while** *not converged and within timeout* **do**
3    **if** *A is feasible* **then**
4      Reduce weights $w_v$ for all nodes to promote cost efficiency
5    **else**
6      Identify memory violation intervals in $A$
7      Increase $w_v$ for nodes active in violated intervals
8    **end**
9    Solve relaxed CFN problem using updated weights to obtain assignment $A$
10 **end**

---

relaxed objective with the true feasibility region, converging to high-quality solutions that strictly respect memory constraints. To guarantee a valid starting point, the penalty weights $w_v$ are initialized such that solving the relaxed CFN yields a feasible assignment $A$, ensuring that the solver never returns an invalid solution. In practice, multiple solver threads are launched in parallel with globally uniform weights sampled on a logarithmic scale (for example 0.1, 1, 10, 100) and small seed-based random perturbations are applied to enhance diversity between runs. The logarithmic initialization makes it very likely that at least one thread finds a feasible solution in the first attempt, as the penalty quickly dominates the original strategy costs. However, if no valid solution is found the initialization continues on the logarithmic scale (for example $10^3$, $10^4$, $10^5$, $10^6$). The weights of the lowest-cost feasible solution are then used as the initial $w_v$. When a feasible solution is obtained, penalties are reduced for all nodes to promote cost efficiency. When violations occur, penalties are increased for nodes active in overloaded intervals to restore feasibility. Through this adaptive reweighting, the solver balances feasibility and efficiency, ensuring robustness and fast convergence even on large graphs under tight time budgets.

**Post-processing.** Solutions computed during the execution of Algorithm 1, which repeatedly solves the relaxed CFN problem, correspond to a surrogate objective rather than the original constrained optimization problem. Although these solutions are often feasible and of low cost, they can frequently be further improved. To address this, we apply a greedy post-processing step. Given a feasible assignment from the relaxed problem, this refinement procedure explores local strategy swaps at individual nodes that reduce the original objective while maintaining feasibility with respect to the memory constraints. Using the example from Figure 1, Figure 3 illustrates a single local swap that reduces the total cost. Note that to perform a local, node-based strategy swap, it is sufficient to consider only the memory usage, the unary cost of the strategy itself, and the binary edge costs between the node and its immediate neighbors (i.e., predecessors and successors in the graph). This localized scope allows for efficient evaluation of potential improvements without requiring recomputation of the full objective.

## 5 EVALUATION

In this section, we first present the benchmark and official contest results, highlighting the performance of our solver in comparison to other submissions. Second, we compare our solver against XLA on all benchmark instances. Third, we evaluate the quality of the solutions produced by our solver

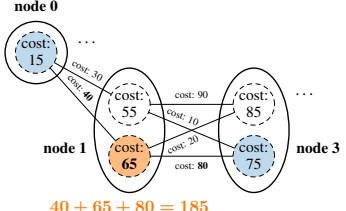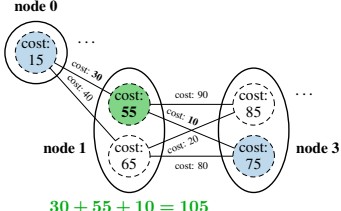

Figure 3: Local strategy swap. In the initial assignment (left), node 1 selects its second strategy. Switching to the first strategy (right) lowers the total cost while preserving feasibility under memory constraints.

by comparing them to lower bounds where available. Finally, we analyze the solver's convergence behavior, focusing on how quickly it reaches high-quality solutions. The hardware used for both the official contest evaluation and the XLA measurements was a Linux virtual machine with an AMD EPYC 7B12 processor, limited to 8 cores and 32 GB of RAM. The lower-bound and convergence evaluations were conducted on nodes equipped with two Intel Xeon Gold 6140 CPUs (18 cores each, 2.3 GHz) and 192 GB of RAM. However, in line with the competition rules, we restricted our jobs to 8 cores and 32 GB of RAM. As CFN solver, we use toulbar2 (Allouche et al., 2015; Trösser et al., 2020; Montalbano et al., 2022; Toulbar2 Developers, 2024) which is capable of handling large-scale instances involving tens of thousands of variables efficiently. Our solver code and evaluation scripts are included in the supplementary materials and will be released upon acceptance.

## 5.1 BENCHMARK INSTANCES

Beyond their role in the contest, the benchmark suite itself is of independent interest. Curated by Google from real production workloads and publicly available (Moffitt & Fegade, 2025a), it spans a broad range of neural architectures, including Graph Network Simulators, U-Nets for vision, diffusion models for generative tasks, Gemma language models, and Transformers. Both supervised fine-tuning and inference tasks are represented. Each benchmark preserves the authentic computation graph structure, candidate execution strategies, and time-varying memory usage profiles found in production systems, ensuring that optimization results translate directly to real-world deep learning workloads.

To support solver development, the contest organizers released five public benchmark instances, each modeling a real-world machine learning workload with varying graph sizes and time constraints. These examples served as test cases for participants to design and validate their optimization strategies. The remaining 20 benchmark instances were kept hidden and used exclusively for final evaluation. Table 1 summarizes the key characteristics of the publicly available problems, including the number of nodes, edges, average strategies per node, file size, and the allowed timeout for computing a valid strategy assignment. An extended table for all 25 benchmark instances, including the hidden ones, is included in Appendix A.

Table 1: Public benchmark instances released by the contest organizers.

| Benchmark name | # Nodes | # Edges | Avg. strat. per node | File size | Timeout |
|---|---|---|---|---|---|
| asplos-2025-iopddl-A | 34,932 | 54,801 | 6,119 | 90M | 60 seconds |
| asplos-2025-iopddl-G | 816 | 1,023 | 12,342 | 2.4M | 120 seconds |
| asplos-2025-iopddl-M | 32,894 | 47,067 | 8,087 | 67M | 180 seconds |
| asplos-2025-iopddl-S | 28,526 | 38,826 | 8,686 | 57M | 240 seconds |
| asplos-2025-iopddl-Y | 62,185 | 91,020 | 20,248 | 1.3G | 300 seconds |

## 5.2 OFFICIAL CONTEST RESULTS

In total, twenty teams submitted a functional solver. The contest organizers evaluated each submission on twenty withheld real-world benchmark instances (Moffitt & Fegade, 2025a). A detailed description of all benchmark characteristics and instances is provided in Appendix A. Each team's solver was

executed under a strict timeout constraint specific to each benchmark instance. Scoring was based on cost minimization. For each benchmark instance, the total cost of a team's solution was compared against the best cost achieved by any team on that instance. The normalized score for a benchmark instance was computed as $\text{score}_{t,b} = \text{min\_cost}_b \,/\, \text{cost}_{t,b}$, where $t$ is the team and $b$ is the benchmark instance. A higher score reflects a lower cost. The overall team score was the sum of its normalized scores across all benchmarks. A detailed example illustrating how the evaluation scores were computed is provided in Appendix B. Note that lower-cost solutions generally translate into shorter step times during training and inference, and as such directly improve overall system efficiency and scalability.

The evaluation across the 20 hidden benchmark instances resulted in a distribution of normalized scores as shown in Figure 4. The first-place submission corresponds to the solution presented in this paper. The individual total scores of the top six ranked teams were as follows: 18.74, 13.84, 13.19, 9.31, 7.92, and 6.56.

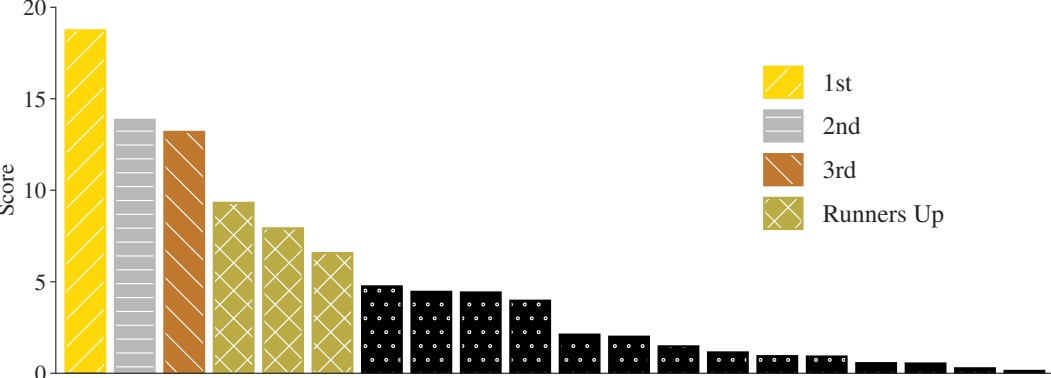

Figure 4: Final scores of all teams on the 20 hidden benchmark instances.

While the score of 18.74 was sufficient to secure first place, closer inspection of the results revealed suboptimal performance on two specific benchmarks: W and V. These were caused by bugs in the solver, which have since been corrected. After addressing these issues, the improved version of the solver achieves an estimated score approaching the theoretical maximum of 20. Appendix C contains detailed evaluation results of the solver on all 25 benchmark instances, including comparisons across the top six ranked teams.

### 5.3 COMPARISON TO XLA

Thanks to support from Google, we obtained official evaluation results for all 25 benchmark instances using XLA (XLA Developers, 2025), the production-grade compiler employed in TensorFlow (Abadi et al., 2016) and JAX (Bradbury et al., 2018), executed on the same contest hardware. Figure 5 shows normalized scores comparing our fixed solver variant to XLA and to the second- and third-place contest submissions. Scores are capped at 1.0 and represent the ratio of the best-known cost to that of each method. XLA consistently underperforms on most benchmarks and fails entirely on challenging cases such as W and X. In contrast, our solver achieves top scores across the full benchmark suite, including on instances like V and W, where the original contest version had previously struggled.

### 5.4 DEVIATIONS FROM LOWER BOUNDS

The competition results demonstrate that our solver performs best relative to other teams and to XLA (XLA Developers, 2025). To contextualize these results more rigorously, we compare the costs of our solutions, obtained under competition timeouts of at most 5 minutes, with lower bounds computed using Gurobi (Gurobi Optimization, LLC, 2025) and Lagrangian relaxation. We selected six instances, representing a diverse set of base models, Graph Network Simulator (GNS), U-Net (UN), Gemma 1 (G1), and Gemma 2 (G2), from the competition benchmark. For each of these instances, Gurobi was run for up to 72 hours. Additionally, we used toulbar2 to obtain lower bounds via the Lagrangian dual and took the best lower bounds found by any method. Although it

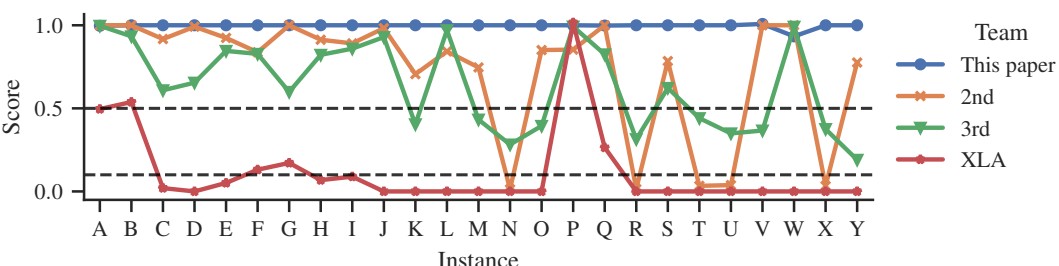

Figure 5: Normalized scores for all 25 benchmark instances comparing the fixed version of our solver to the second and third place teams and to XLA. A higher score indicates a lower cost relative to the best solution found for that instance.

remains unclear whether the lower bounds are achievable in general, our solver produces solutions close to these bounds. Table 5 in the Appendix presents a detailed comparison of our solver's costs against the computed lower bounds for all benchmark instances. The final row of the table summarizes the relative gap, defined as the ratio between our solver's cost and the lower bound. Across all instances, our solver's solutions are within 27% of the computed lower bounds, with many instances exhibiting gaps below 5%. This indicates that our solver not only outperforms other methods but also approaches near-optimal solutions as indicated by the lower bounds.

Table 2: Comparison of our solver's total cost to lower bounds (LB). The last row reports the relative gap, i.e., the ratio between our solver's cost and the lower bound.

| Instance (Model) | A (GNS) | K (GNS) | L (UN) | M (G1) | R (G2) | S (UN) |
|---|---|---|---|---|---|---|
| **This paper** | 6.57e+09 | 2.27e+09 | 1.97e+09 | 4.99e+10 | 3.72e+11 | 1.53e+09 |
| **LB** | 6.51e+09 | 2.07e+09 | 1.92e+09 | 4.94e+10 | 3.05e+11 | 1.51e+09 |
| **Cost / LB** | 1.01 | 1.10 | 1.03 | 1.01 | 1.22 | 1.02 |

## 5.5 SOLVER CONVERGENCE

In this subsection, we analyze the convergence behavior of our solver for the instances of the previous subsection. Figure 6 shows the cost trajectory over time, where the $y$-axis indicates the relative cost compared to the best solution. For instances A, L, and M, the solver converges within just a few seconds. The remaining instances require several iterations to reach their best results. Nevertheless, even for these more challenging cases, the initial solution is already close in quality to the final one. The first result found by our solver is with uniform weights, on easier instances these seem to suffice, while on harder instances individual node weights lead to up to 2 times better solutions.

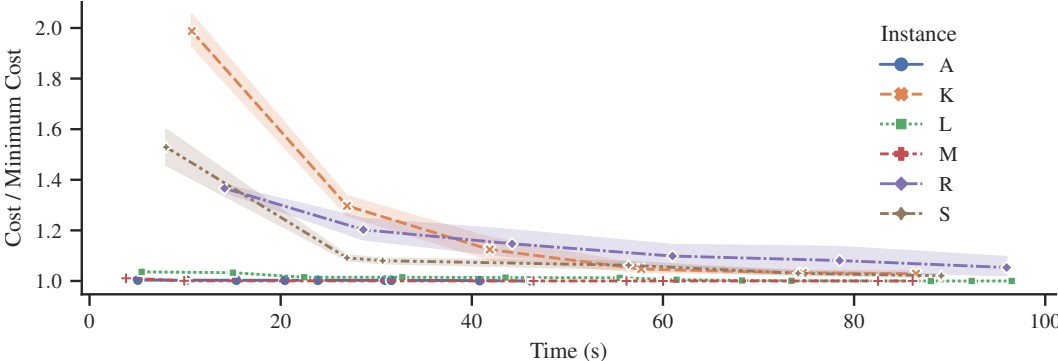

Figure 6: Convergence behavior of our solver on six representative benchmark instances. The $y$-axis shows the relative cost (i.e., current cost divided by the best cost found), and the x-axis represents time. Shaded regions indicate confidence intervals over 10 random seeds.

The fast convergence behavior is not limited to the representative instances shown in Figure 6. Our solver shows similarly rapid progress toward low-cost solutions more generally, as illustrated by the convergence plot for the four largest benchmark instances included in Appendix D.

## 6 DISCUSSION

The results of our solver in the ASPLOS / EuroSys 2025 contest highlight the effectiveness of usage-constrained relaxation as a strategy for solving large-scale, constraint-heavy optimization problems in distributed deep learning. By converting strict memory constraints into soft penalties and tuning them adaptively, we enable flexible exploration of the solution space without compromising feasibility or quality.

Our formulation also lends itself well to integration into modern compiler infrastructures such as Alpa (Zheng et al., 2022), XLA (XLA Developers, 2025), or TVM (Chen et al., 2018). Because memory usage is incorporated directly into the cost function, the method can be combined with multi-objective optimizers or extended to other resource dimensions such as bandwidth or power.

The ability to efficiently compute high-quality intra-operator strategies further benefits higher-level compiler decisions. Systems for inter-operator parallelism, such as pipelining or stage partitioning, must often assume fixed intra-operator costs or rely on coarse approximations. Our solver provides realistic estimates fast enough to embed directly into inter-operator search, enabling more informed pipeline-level decisions and better end-to-end performance across heterogeneous device meshes.

## 7 CONCLUSIONS

We presented a scalable solution to intra-operator parallelism based on usage-constrained relaxation, which integrates memory constraints into the cost model via adaptive penalties. Despite its simplicity, this approach enables our solver to handle large graphs efficiently and consistently produces valid, low-cost solutions within strict time budgets. By combining cost function networks, adaptive weight tuning, and greedy refinement, the solver achieves state-of-the-art performance on real-world benchmark problems. Compared to XLA, our solver consistently produces solutions that are often orders of magnitude lower in cost, particularly on large problem instances. This will result in faster execution times of deep learning workloads. Looking ahead, this relaxation-based strategy, combined with adaptive per-node weight adjustments, provides a general framework for tackling other resource-constrained optimization problems in deep learning systems.

## REPRODUCIBILITY STATEMENT

The source code of the solver and all experiments is included in the supplementary materials. The main steps needed to reproduce the experiments are as follows:

1. Create a Python environment with the required dependencies. This is best done with uv. Installation instructions are available at `https://docs.astral.sh/uv/getting-started/installation/`. Any other compatible package manager will also work.

2. Download the benchmark instances from `https://github.com/google/iopddl/tree/main/benchmarks`. A script to download and unpack them is included in the top level `README.md`.

3. Each experiment is placed in its own folder and includes a script for execution. More details can be found in the top level `README.md`.

The experiments include a prebuilt static binary of the solver that works on 64-bit x86 Linux-based systems. For convenience, we also provide a Dockerfile that builds and runs the solver and experiments on any platform that supports Docker.

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

# A    BENCHMARK INSTANCES

The benchmark suite captures a wide spectrum of scheduling and memory optimization challenges encountered in realistic distributed deep learning workloads. Figure 7 provides an overview of all 25 benchmark instances used in the ASPLOS / EuroSys 2025 contest on Intra-Operator Parallelism (Moffitt & Fegade, 2025a). The table summarizes key characteristics for each instance, including:

**Base Model:** The underlying neural architecture, spanning Graph Network Simulator (GNS), U-Net (UN), Gemma 1 (G1), Gemma 2 (G2), Transformer, and Diffusion models.

**Task:** Either Supervised Fine-Tuning (SFT) or Inference.

**Device Mesh:** The logical layout of devices available for parallelization, impacting strategy granularity and communication.

**Crosscuts:** Whether artificial crosscutting edges have been added to enforce identical strategies for structurally duplicated nodes (e.g., from loop unrolling).

**∞-elim:** Whether infeasible (infinite-cost) strategy combinations were removed during preprocessing.

**#Nodes / #Edges:** Size of the computational graph.

**Avg. Strat.:** The average number of viable strategies per node, indicating the branching factor of the search space.

**Filesize:** Size of the serialized benchmark graph.

**Timeout:** Time limit used during evaluation.

**Subset:** Indicates whether the instance was part of the public or private set. Public instances were available for development, while private ones were withheld until the final evaluation.

| Benchmark Name | Base Model | Task | Device Mesh | Crosscuts? | ∞-elim? | #Nodes | #Edges | Avg. Strat. | Filesize | Timeout | Subset |
|---|---|---|---|---|---|---|---|---|---|---|---|
| asplos-2025-iopddl-A | GNS | SFT | [4, 8] | ✓ | | 34,932 | 54,801 | 6.119 | 90M | 60 sec. | public |
| asplos-2025-iopddl-B | Transformer | Infer. | [4, 8] | ✓ | | 816 | 1,096 | 12.485 | 3.0M | 60 sec. | private |
| asplos-2025-iopddl-C | Gemma 1 (2B) | SFT | [8, 8] | ✓ | | 32,894 | 52,234 | 8.087 | 83M | 60 sec. | private |
| asplos-2025-iopddl-D | Gemma 1 (7B) | SFT | [8, 8] | ✓ | | 40,958 | 68,635 | 8.792 | 139M | 60 sec. | private |
| asplos-2025-iopddl-E | Gemma 1 (9B) | SFT | [8, 8] | ✓ | | 59,711 | 93,338 | 9.630 | 226M | 60 sec. | private |
| asplos-2025-iopddl-F | Gemma 1 (27B) | SFT | [8, 16] | ✓ | | 65,335 | 102,613 | 9.619 | 249M | 120 sec. | private |
| asplos-2025-iopddl-G | Transformer | Infer. | [2, 16] | | | 816 | 1,023 | 12.342 | 2.4M | 120 sec. | public |
| asplos-2025-iopddl-H | Gemma 2 (9B) | SFT | [8, 8] | ✓ | | 56,833 | 91,053 | 9.291 | 210M | 120 sec. | private |
| asplos-2025-iopddl-I | Gemma 2 (27B) | SFT | [8, 16] | ✓ | | 62,185 | 100,112 | 9.284 | 232M | 120 sec. | private |
| asplos-2025-iopddl-J | Diffusion | SFT | [2, 16] | ✓ | | 60,206 | 102,187 | 9.050 | 166M | 120 sec. | private |
| asplos-2025-iopddl-K | GNS | SFT | [2, 16] | | | 34,932 | 49,674 | 6.122 | 28M | 180 sec. | private |
| asplos-2025-iopddl-L | U-Net | SFT | [4, 8] | ✓ | | 28,526 | 44,512 | 9.085 | 95M | 180 sec. | private |
| asplos-2025-iopddl-M | Gemma 1 (2B) | SFT | [8, 8] | | | 32,894 | 47,067 | 8.087 | 67M | 180 sec. | public |
| asplos-2025-iopddl-N | Gemma 1 (7B) | SFT | [8, 8] | | ✓ | 40,958 | 41,606 | 8.223 | 31M | 180 sec. | private |
| asplos-2025-iopddl-O | Gemma 1 (9B) | SFT | [8, 8] | | | 59,711 | 83,862 | 9.630 | 177M | 180 sec. | private |
| asplos-2025-iopddl-P | Gemma 1 (27B) | SFT | [8, 16] @0 | ✓ | | 65,335 | 100,547 | 2.796 | 16M | 240 sec. | private |
| asplos-2025-iopddl-Q | Gemma 2 (9B) | SFT | [8, 8] @1 | ✓ | | 56,833 | 91,053 | 7.456 | 114M | 240 sec. | private |
| asplos-2025-iopddl-R | Gemma 2 (27B) | SFT | [8, 16] | | ✓ | 62,185 | 54,990 | 8.637 | 48M | 240 sec. | private |
| asplos-2025-iopddl-S | U-Net | SFT | [2, 16] | | | 28,526 | 38,826 | 8.686 | 57M | 240 sec. | public |
| asplos-2025-iopddl-T | Diffusion | SFT | [4, 8] | | ✓ | 60,206 | 41,764 | 8.940 | 44M | 240 sec. | private |
| asplos-2025-iopddl-U | GNS | SFT | [2, 4, 4] | | ✓ | 34,932 | 20,188 | 8.079 | 26M | 300 sec. | private |
| asplos-2025-iopddl-V | Transformer | Infer. | [2, 4, 4] | | | 816 | 1,023 | 29.086 | 18M | 300 sec. | private |
| asplos-2025-iopddl-W | Diffusion | SFT | [2, 4, 4] | ✓ | | 60,206 | 101,975 | 16.030 | 1.1G | 300 sec. | private |
| asplos-2025-iopddl-X | Gemma 1 (9B) | SFT | [4, 4, 4] | | ✓ | 59,711 | 50,755 | 19.240 | 262M | 300 sec. | private |
| asplos-2025-iopddl-Y | Gemma 2 (27B) | SFT | [4, 4, 8] | | | 62,185 | 91,020 | 20.248 | 1.3G | 300 sec. | public |

Figure 7: Overview of all 25 benchmark instances from the ASPLOS / EuroSys 2025 contest. Each row corresponds to a different instance with detailed metadata including model type, graph structure, strategy complexity, and resource constraints.

# B    COMPUTATION OF NORMALIZED SCORES

To illustrate how evaluation scores were computed, Table 3 shows a hypothetical example involving three benchmark instances. For each instance, the score of a team is calculated by dividing the minimum cost achieved on that instance by the team's cost. The total score is the sum of these

normalized scores across all benchmarks. In this example, Team A achieves the highest total score and is therefore ranked first.

Table 3: Example evaluation: raw costs, normalized scores, and total rankings across three fictitious benchmark instances X, Y, and Z.

| Team | Raw cost | | | Score | | | Total | Rank |
|---|---|---|---|---|---|---|---|---|
| | X | Y | Z | X | Y | Z | Score | |
| Team A | 100 | 500 | 800 | 1.000 | 0.400 | 0.500 | **1.900** | 1 |
| Team B | 300 | 200 | 900 | 0.333 | 1.000 | 0.444 | **1.778** | 2 |
| Team C | 600 | 700 | 400 | 0.167 | 0.286 | 1.000 | **1.452** | 3 |
| **Min. Cost** | 100 | 200 | 400 | | | | | |

## C    FULL COMPETITION RESULTS OF THE TOP SIX TEAMS

Table 4 presents the full cost and score results across all benchmark instances for the top six teams in the ASPLOS / EuroSys 2025 contest. For each instance, we report the cost and corresponding normalized score achieved by our solver, as well as by the second through sixth place teams. Additionally, the final column shows the cost produced by the XLA (XLA Developers, 2025) compiler for comparison. The lowest (i.e., best) cost for each benchmark instance is highlighted in bold.

Table 4: Official contest results on all 25 benchmark instances.

| | Instance | This paper | 2nd place | 3rd place | 4th place | 5th place | 6th place | XLA |
|---|---|---|---|---|---|---|---|---|
| A | Cost | 6.57e+09 | **6.55e+09** | 6.57e+09 | 1.40e+10 | 6.87e+09 | 2.29e+10 | 1.32e+10 |
| | Score | 1.00 | 1.00 | 1.00 | 0.47 | 0.95 | 0.29 | 0.50 |
| B | Cost | **5.33e+05** | 5.33e+05 | 5.71e+05 | 1.25e+07 | 9.08e+06 | 1.19e+08 | 9.90e+05 |
| | Score | 1.00 | 1.00 | 0.93 | 0.04 | 0.06 | 0.00 | 0.54 |
| C | Cost | **8.41e+10** | 9.17e+10 | 1.38e+11 | 3.56e+11 | 4.54e+11 | 2.24e+12 | 4.20e+12 |
| | Score | 1.00 | 0.92 | 0.61 | 0.24 | 0.19 | 0.04 | 0.02 |
| D | Cost | **3.14e+11** | 3.16e+11 | 4.79e+11 | 6.22e+11 | 2.00e+18 | 1.52e+12 | N/A |
| | Score | 1.00 | 0.99 | 0.65 | 0.50 | 0.00 | 0.21 | N/A |
| E | Cost | **3.39e+11** | 3.67e+11 | 4.01e+11 | 4.48e+11 | 2.00e+18 | 5.74e+11 | 6.64e+12 |
| | Score | 1.00 | 0.92 | 0.85 | 0.76 | 0.00 | 0.59 | 0.05 |
| F | Cost | **3.64e+11** | 4.35e+11 | 4.40e+11 | 6.63e+11 | 2.00e+18 | 9.39e+11 | 2.79e+12 |
| | Score | 1.00 | 0.84 | 0.83 | 0.55 | 0.00 | 0.39 | 0.13 |
| G | Cost | **2.17e+05** | 2.17e+05 | 3.62e+05 | 7.26e+06 | 1.18e+06 | 5.58e+07 | 1.26e+06 |
| | Score | 1.00 | 1.00 | 0.60 | 0.03 | 0.18 | 0.00 | 0.17 |
| H | Cost | **5.28e+11** | 5.78e+11 | 6.42e+11 | 6.87e+11 | 2.00e+18 | 8.55e+11 | 7.75e+12 |
| | Score | 1.00 | 0.91 | 0.82 | 0.77 | 0.00 | 0.62 | 0.07 |
| I | Cost | **5.84e+11** | 6.55e+11 | 6.79e+11 | 1.03e+12 | 2.00e+18 | 1.50e+12 | 6.55e+12 |
| | Score | 1.00 | 0.89 | 0.86 | 0.56 | 0.00 | 0.39 | 0.09 |
| J | Cost | **1.36e+12** | 1.38e+12 | 1.47e+12 | 6.03e+12 | 1.46e+12 | 6.34e+20 | N/A |
| | Score | 1.00 | 0.98 | 0.93 | 0.23 | 0.93 | 0.00 | N/A |
| K | Cost | **2.27e+09** | 3.22e+09 | 5.63e+09 | 1.23e+10 | 5.01e+09 | 2.03e+10 | N/A |
| | Score | 1.00 | 0.71 | 0.40 | 0.18 | 0.45 | 0.11 | N/A |
| L | Cost | **1.97e+09** | 2.34e+09 | 2.03e+09 | 7.57e+09 | 2.15e+09 | 7.29e+09 | N/A |
| | Score | 1.00 | 0.84 | 0.97 | 0.26 | 0.92 | 0.27 | N/A |
| M | Cost | **4.99e+10** | 6.69e+10 | 1.15e+11 | 3.25e+11 | 7.46e+10 | 8.36e+11 | N/A |
| | Score | 1.00 | 0.75 | 0.43 | 0.15 | 0.67 | 0.06 | N/A |
| N | Cost | **2.03e+11** | 1.50e+13 | 7.18e+11 | 6.06e+11 | 5.87e+11 | 1.33e+12 | N/A |
| | Score | 1.00 | 0.01 | 0.28 | 0.33 | 0.35 | 0.15 | N/A |

| | Instance | This paper | 2nd place | 3rd place | 4th place | 5th place | 6th place | XLA |
|---|---|---|---|---|---|---|---|---|
| O | Cost | **2.23e+11** | 2.62e+11 | 5.65e+11 | 3.80e+11 | 5.12e+11 | 5.18e+11 | N/A |
| | Score | 1.00 | 0.85 | 0.39 | 0.59 | 0.43 | 0.43 | N/A |
| P | Cost | 6.62e+11 | 7.76e+11 | 6.67e+11 | 9.40e+11 | 1.21e+12 | 9.53e+11 | **6.52e+11** |
| | Score | 0.98 | 0.84 | 0.98 | 0.69 | 0.54 | 0.68 | 1.00 |
| Q | Cost | 5.28e+11 | **5.27e+11** | 6.39e+11 | 6.63e+11 | 1.35e+12 | 8.76e+11 | 2.00e+12 |
| | Score | 1.00 | 1.00 | 0.82 | 0.79 | 0.39 | 0.60 | 0.26 |
| R | Cost | **3.72e+11** | 2.47e+13 | 1.18e+12 | 7.47e+11 | 6.02e+11 | 1.53e+12 | N/A |
| | Score | 1.00 | 0.02 | 0.31 | 0.50 | 0.62 | 0.24 | N/A |
| S | Cost | **1.53e+09** | 1.96e+09 | 2.46e+09 | 3.31e+09 | 2.43e+09 | 8.58e+09 | N/A |
| | Score | 1.00 | 0.78 | 0.62 | 0.46 | 0.63 | 0.18 | N/A |
| T | Cost | **6.83e+11** | 2.03e+13 | 1.55e+12 | 1.08e+12 | 7.80e+11 | 1.45e+12 | N/A |
| | Score | 1.00 | 0.03 | 0.44 | 0.63 | 0.88 | 0.47 | N/A |
| U | Cost | **2.57e+09** | 6.75e+10 | 7.36e+09 | 6.60e+09 | 5.06e+09 | 8.70e+09 | N/A |
| | Score | 1.00 | 0.04 | 0.35 | 0.39 | 0.51 | 0.30 | N/A |
| V | Cost | 2.21e+06 | **1.63e+06** | 4.45e+06 | 6.25e+07 | 9.54e+06 | 2.51e+08 | N/A |
| | Score | 0.74 | 1.00 | 0.37 | 0.03 | 0.17 | 0.01 | N/A |
| W | Cost | 9.04e+20 | **1.18e+13** | 1.19e+13 | 1.85e+13 | 1.65e+13 | 2.25e+13 | N/A |
| | Score | 0.00 | 1.00 | 0.99 | 0.64 | 0.72 | 0.52 | N/A |
| X | Cost | **2.80e+11** | 8.39e+12 | 7.47e+11 | 4.55e+11 | 3.68e+11 | 5.36e+11 | N/A |
| | Score | 1.00 | 0.03 | 0.38 | 0.62 | 0.76 | 0.52 | N/A |
| Y | Cost | **6.21e+11** | 8.01e+11 | 3.26e+12 | 1.62e+12 | 1.09e+12 | 1.47e+12 | N/A |
| | Score | 1.00 | 0.78 | 0.19 | 0.38 | 0.57 | 0.42 | N/A |

## D  CONVERGENCE RESULTS ON THE LARGEST INSTANCES

In the main paper, we demonstrated strong convergence behavior on benchmark instances for which we were able to compute lower bounds using `Gurobi` (Gurobi Optimization, LLC, 2025). To further substantiate the robustness of our approach, we now examine convergence behavior on the four largest benchmark instances: `F`, `W`, `X`, and `Y`. Figure 8 shows the relative cost over time, measured against the best solution found. Despite the increased complexity and scale of these benchmark instances, our solver quickly converges to high-quality solutions.

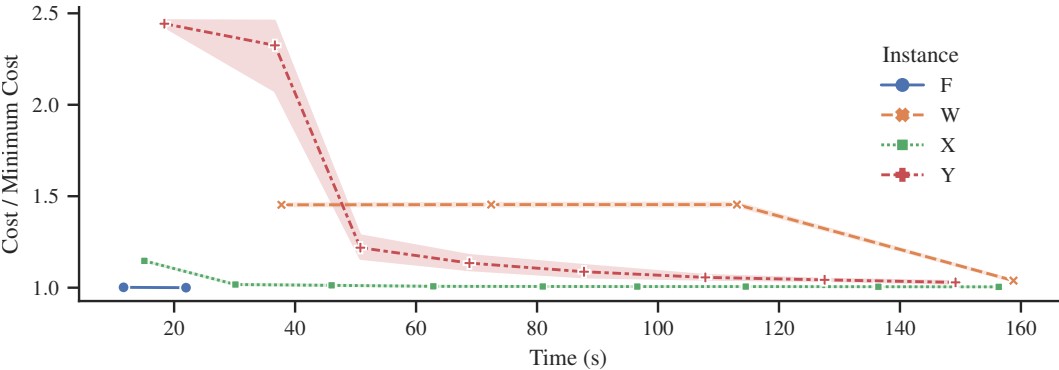

Figure 8: Convergence behavior of our solver on the four largest benchmark instances. The y-axis shows the relative cost (i.e., current cost divided by the best cost found), and the x-axis represents time. Shaded regions indicate confidence intervals over 10 random seeds.

## E  ABLATION STUDY ON GREEDY POST-PROCESSING

In this section, we present an ablation study of the greedy post-processing step in our solver, which is applied after finding a feasible usage-relaxed solution. We compare solver performance with and without this step by calculating the *reduction percentage*, defined as $(\text{cost}_{\text{before}} - \text{cost}_{\text{after}}) / \text{cost}_{\text{before}}$.

The results for the six representative benchmark instances used in the main paper are shown in Figure 9. Greedy post-processing is most beneficial early in the optimization process, when the weights are not yet well tuned. This is intuitive, as such solutions leave more room for further improvement. On instances that quickly converge near the optimum, such as A, L, and M, the effect of post-processing is negligible.

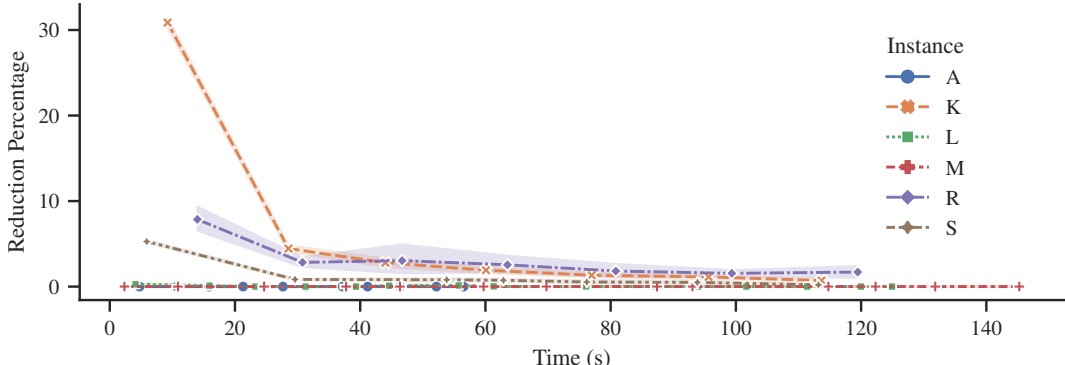

Figure 9: Reduction percentage of the greedy post-processing on the six representative instances. Shaded regions indicate confidence intervals over 10 random seeds.

We ran the same experiment on the four largest benchmark instances: F, W, X, and Y. The results are shown in Figure 10. While greedy post-processing is less effective on these larger instances, it still provides benefits, particularly when the solver struggles to find good solutions early on, similar to the behavior observed in the previous six benchmark instances.

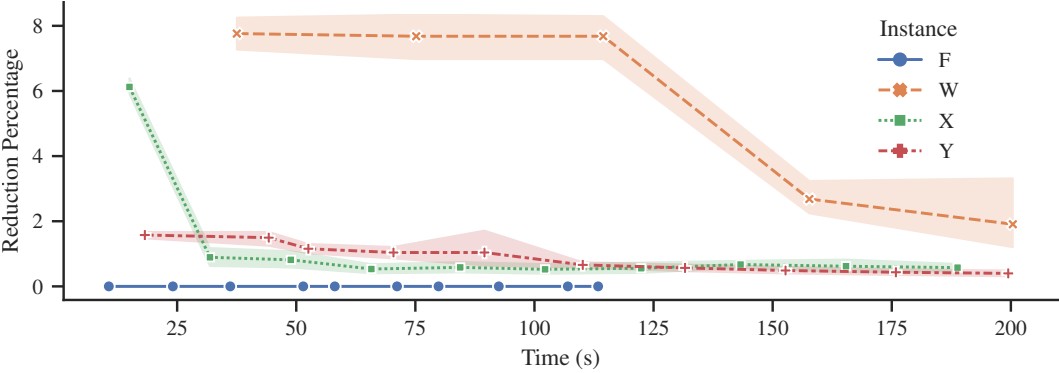

Figure 10: Reduction percentage of the greedy post-processing on the four largest instances. Shaded regions indicate confidence intervals over 10 random seeds.

## F LLM USAGE

Large Language Models (LLMs) were used as supportive tools during the preparation of this paper. They assisted in refining the writing style of individual sentences, suggesting alternative phrasings for clarity, and improving the readability of technical explanations. LLMs were also consulted for feedback on figure captions, aesthetics, and layout suggestions. They were not involved in developing research ideas, designing methods, conducting experiments, or analyzing results. All technical contributions and findings were produced independently by the authors and carefully verified. The authors take full responsibility for the accuracy and originality of the final paper.

# G LOWER BOUNDS

Our solver consistently finds high-quality solutions across all benchmark instances. To quantify how close these solutions are to the optimal, we computed lower bounds using `Gurobi` and `toulbar2` on all instances. Table 5 summarizes the results, showing the best cost found by our solver, the computed lower bound, and the resulting optimality gap for each instance. The gaps are generally small, indicating that our solver's solutions are near-optimal.

Table 5: Comparison of our solver's total cost to lower bounds (LB) computed with `Gurobi`. The last row reports the relative gap, i.e., the ratio between our solver's cost and the lower bound.

| This paper | LB | Cost / LB |
|---|---|---|
| 6.57e+09 | 6.51e+09 | 1.01 |
| 5.33e+05 | 5.33e+05 | 1.00 |
| 8.41e+10 | 8.25e+10 | 1.02 |
| 3.14e+11 | 2.98e+11 | 1.05 |
| 3.39e+11 | 3.23e+11 | 1.05 |
| 3.64e+11 | 3.54e+11 | 1.03 |
| 2.17e+05 | 217039 | 1.00 |
| 5.28e+11 | 5.08e+11 | 1.04 |
| 5.84e+11 | 5.57e+11 | 1.05 |
| 1.36e+12 | 1.11e+12 | 1.22 |
| 2.27e+09 | 2.07e+09 | 1.10 |
| 1.97e+09 | 1.92e+09 | 1.03 |
| 4.99e+10 | 4.94e+10 | 1.01 |
| 2.03e+11 | 1.73e+11 | 1.17 |
| 2.23e+11 | 1.91e+11 | 1.17 |
| 6.62e+11 | 6.16e+11 | 1.07 |
| 5.28e+11 | 5.06e+11 | 1.04 |
| 3.72e+11 | 3.05e+11 | 1.22 |
| 1.53e+09 | 1.51e+09 | 1.02 |
| 6.83e+11 | 5.86e+11 | 1.17 |
| 2.57e+09 | 2.02e+09 | 1.27 |
| 1.62e+06 | 1.62e+06 | 1.00 |
| 1.26e+13 | 1.07e+13 | 1.18 |
| 2.80e+11 | 2.45e+11 | 1.15 |
| 6.21e+11 | 4.95e+11 | 1.25 |

