# OpenReview forum: "Soft Constraints, Strong Solutions: Optimizing Intra-Operator Parallelism for Distributed Deep Learning"
_ICLR.cc/2026/Conference — Submitted to ICLR 2026_

### Official Review · Reviewer_i5Lc · 2025-10-24

**Soundness:** 3
**Presentation:** 3
**Contribution:** 3
**Rating:** 6
**Confidence:** 3

**Summary:**

The authors propose an approximate algorithm for a combinatorial optimization problem derived from intra-operator parallelization in distributed LLM training and inference. Given a graph where each node has an activation time interval and a set of strategies (each with memory usage and time cost), and where each edge's time cost depends on the strategies chosen at its endpoints, the goal is to assign a strategy to every node to minimize total time subject to a memory budget. The proposed algorithm won an ASPLOS contest for this problem. Experiments show the algorithm typically finds solutions with latency between 1× and 2× the optimal.

**Strengths:**

1. The algorithm is simple yet effective.
2. The optimization problem comes from a real deployment scenario, so the work has practical impact.
3. The algorithm outperforms other contest teams and the production compiler XLA.

**Weaknesses:**

1. The paper would benefit from a clearer connection between the formal optimization problem and the original intra-operator parallelization task in LLM training/serving.
2. It is surprising that a well-optimized system like XLA can be an order of magnitude slower in some cases; the paper should provide more explanation for this gap.

**Questions:**

Thanks for submitting to ICLR 2026! I enjoyed reading the paper; it is well written and easy to follow. The algorithm is simple and effective. I have a few suggestions to improve clarity and impact.

**1. Reconnect the optimization problem to intra-operator parallelization.**

The ASPLOS contest abstracts a production problem into a combinatorial formulation, but the paper should better describe the original system-level problem so the paper is self-contained. This will help readers map the mathematical solution back to a concrete scheduling or parallelization strategy in production.

**2. Explain differences with XLA's schedule.**

If the reported cost measures execution latency of the computation graph under a strategy assignment, it is surprising that XLA’s approach can be much slower. An in-depth comparison (or diagnosis) explaining why your solution improves over XLA, for example, differences in objective, search space, heuristics, or memory-time trade-offs, would strengthen the paper.

Overall, this is a solid and practical contribution. Addressing the two points above will make the paper more useful to practitioners and clearer to reviewers.

---

> ### Author Response · Authors · 2025-11-21
>
> Thank you for the detailed review and positive assessment. We address your suggestions below.
>
> 1. Reconnect the optimization problem to intra-operator parallelization.
>
> As this seemed unclear to other reviewers as well, this is an excellent suggestions and we adjust the paper accordingly. An updated version will be uploaded in the next days.
>
> 2. Explain differences with XLA's schedule.
>
> The XLA results are directly comparable as they solve the exact same optimization problem and were provided by the contest organizers, XLA developers, on the same hardware.
> XLA currently uses the Google OR-Tools CP-SAT optimizer to solve the optimization problem.
> CP-SAT is an excellent solver, that just won several prices in the MiniZinc Challenge 2025.
> However, adding the necessary memory constraints introduces large sum constraints as thousands of nodes can be active at the same time and their combined memory usage must not exceed the available memory. These constraints are hard for many solvers, including Gurobi, toulbar2 (the solver we used) and CP-SAT. Our methods avoids modeling these constraints entirely by penalizing each node individually. Additionally, the unconstrained version perfectly fits cost function networks as the computational graph structure enables the graph based optimization they are designed for. Together this leads to the exceptional performance of our solver. The organizers did not believe solution on this level were possible before the contest. We clearify this in the updated paper.
>
> On the impact of better solutions we quote the organizers of the competition:
>
> > Lower-cost solutions will roughly translate to higher accelerator performance (i.e., a reduction in step times for the corresponding training / inference workloads). The improvement may not be exactly proportional to cost, but should be directionally correct.

---

> > ### Comment · Reviewer_i5Lc · 2025-11-25
> >
> > Thanks the authors for the response! I keep my positive view on this work and keep the score 6.

---

### Official Review · Reviewer_jAiB · 2025-10-29

**Soundness:** 1
**Presentation:** 2
**Contribution:** 1
**Rating:** 0
**Confidence:** 5

**Summary:**

This paper presents an intra-operator parallelism solution for a contest, working on a solver for a formulated assignment optimization problem. The key idea is to optimize the objective with relaxed constraints first and greedily adjust the assignment later. It is more of a contest report than an academic paper. It makes little academic contribution: the problem is not new; the method is not novel; the experiment is preliminary.

**Strengths:**

1. The proposed method is a top-performing solution to a contest.

**Weaknesses:**

1. The defined problem for intra-operator parallelism is not contributed by the authors, though the problem itself does not contribute novelty and academic significance either.
2. Authors intentionally design the proposed method (e.g., Techniques 1 and 2 in the proposed method) to fit into the provided workload, while not considering a method for real-case workloads.
3. The idea of solving a relaxed-constraint optimization problem and the adopted algorithm are nothing novel.
4. The proposed method is not properly and throughly verified. The experiment tried to evaluate a solution for a large-scale problem using a ``low''-configured server. Evaluations on a certain scale computing cluster with realistic distributed learning workloads can be much more convincing.

**Questions:**

1. I would advise authors to work on realistic system problems on intra-operator parallelism with evaluation on realistic workloads and environments.

---

> ### Author Response · Authors · 2025-11-21
>
> Thank you for reviewing our paper. Unfortunately, there are some inaccuracies in your review that we address below. We hope you take this into account and update your score accordingly.
>
> *Weakness 1: The defined problem for intra-operator parallelism is not contributed by the authors, though the problem itself does not contribute novelty and academic significance either.*
>
> We state, that the problem definition stems from the ASPLOS contest, which clearly proves its industrial and academic significance. The problem had not been solved to a sufficient degree before we introduced our method.
>
> *Weakness 2: Authors intentionally design the proposed method (e.g., Techniques 1 and 2 in the proposed method) to fit into the provided workload, while not considering a method for real-case workloads.*
>
> As noted several times in the paper these instances were derived from real production workloads, curated and released by Google. These cover a wide range of models and use cases, including Graph Network Simulators, Transformers, U-Nets, diffusion models, and Gemma variants, across both training and inference tasks (see Section 3 line 193-199 and Supplement Section 2 for more information).
>
> *Weakness 3: The idea of solving a relaxed-constraint optimization problem and the adopted algorithm are nothing novel.*
>
> As mentioned before the problem has not been solved to a sufficient degree yet, even though it is highly relevant. While relaxation itself is a standard technique, you still need to efficiently solve the relaxed problem. We show that this is possible using cost function networks and propose a non standard method for optimizing the relaxation, which does not rely on hard to obtain lower bounds and forces feasible solutions. Standard Lagrangian relaxation usually leads to infeasible solutions, that are non trivial to fix, . Moreover, it requires strong lower bounds for optimization of the Lagrangian dual, which are much harder to obtain than upper bounds on feasible solutions. We will upload an updated version of the paper that explains this in more detail in the next days. It is worth noting that none of the 19 other competition teams, including those from major research labs and industry, achieved comparable performance.
>
> *Weakness 4: The proposed method is not properly and throughly verified. The experiment tried to evaluate a solution for a large-scale problem using a ``low''-configured server. Evaluations on a certain scale computing cluster with realistic distributed learning workloads can be much more convincing.*
>
> As already noted we optimize realistic distributed deep learning workloads. Interestingly, our work shows that these real world problem instances can be solved using very limited resources.

---

> > ### Comment · Reviewer_jAiB · 2025-11-25
> >
> > The authors' responses confirm my concerns about the flaws in the paper in the following aspects: the importance of the problem, the novelty in the method design, and a proper design of the experiment. It does not meet the standards for publication in a top-tier conference like ICLR. I decide to maintain my negative rating.

---

> > > ### Author Response · Authors · 2025-11-25
> > >
> > > We respectfully disagree with your assessment of the problem importance. It got attention from top universities (UC Berkley, Carnegie Mellon University, Shanghai Jiao Tong University) and leading industry (Google, Amazon, Microsoft) [Refrences from the paper: Alpa (Zheng et al., 2022), nnScaler (Lin et al., 2024)]. Despite that, the problem remained challenging, as written in the official contest report [1]:
> > >
> > > > Although a variety of techniques have been considered (including random sampling, constraint programming, Monte Carlo Tree Search, etc.), it remains an open question which is the most efficient and effective way to produce high-quality model partitionings at compile time.
> > >
> > > Which also highlights its significance again:
> > > > A chief enabler of large-scale deep learning is the distribution of computation across multiple interconnected hardware
> > > accelerators. In order to unlock the maximum possible performance, a compiler must first select a reasonable strategy
> > > to parallelize a model’s operations.
> > >
> > > The benchmark instances stem from various real world deep learning models, and have been extracted from the strategy generation of XLA. Thus, our solver can be directly integrated into the XLA compiler pipeline, to solve real world instances. Therefore, it remains unclear to us why you believe we do not work on realistic data. If you continue to believe so, please provide justified arguments for your claims.
> > >
> > >  The experiments follow the official contest rules and are not made up by us. Since optimizing inter-operator parallelism is often a substep in a larger optimization pipeline and can be called multiple times for a single instance, quick solving times are important.
> > >
> > > We do not have access to a cluster of interesting size (at least 64 GPUs), to run the workloads directly, nor do we believe that that should be a necessary requirement to do research.
> > >
> > >
> > >
> > > [1] Michael D. Moffitt and Pratik Fegade. 2025. The ASPLOS 2025 / EuroSys 2025 Contest on Intra-Operator Parallelism for Distributed Deep Learning. In Proceedings of the 30th ACM International Conference on Architectural Support for Programming Languages and Operating Systems, Volume 3 (ASPLOS '25). Association for Computing Machinery, New York, NY, USA, 5–17. https://doi.org/10.1145/3676642.3736399

---

> > > > ### Comment · Reviewer_jAiB · 2025-11-28
> > > >
> > > > Thanks for the authors' response and claim.
> > > > 1. For the problem the paper is solving, the large overlap in the background and problem definition with the contest statement [1], while not avoidable, is a significant concern. The authors failed to discuss the problem in depth, such as the cause and consequence of the problem under different node scales, distribution of costs, etc. The authors also failed to address the gap between the formulated problem and the real-world testbeds and workloads, which is important for the applicability of the final solution built on this problem formulation. This paper did not provide new insights into an existing problem.
> > > >
> > > > 2. The response of the author did not effectively address my concerns about the method's novelty. I think "this is possible using cost function networks and propose a non-standard method for optimizing the relaxation, which does not rely on hard to obtain lower bounds and forces feasible solutions" is marginal innovation: adapting an existing general approach for a different problem without new insights and findings.
> > > >
> > > > 3. About real-world performance, I believe end-to-end evaluation is basic and important for algorithms that solve a system problem, especially for one that claims to be effective in distributed training that requires a certain scale, as in the studies the authors cited did [Alpa (Zheng et al., 2022), nnScaler (Lin et al., 2024)]. Though the evaluation on a generated workload profile can provide preliminary evidence of the proposed method, it can be far from enough to verify the end-to-end performance when running in real machines, where the dynamics of hardware affect the final performance significantly. While "we do not have access to a cluster of interesting size (at least 64 GPUs)" can be a reason for not being able to conduct the end-to-end performance on a large-scale cluster, it is not an excuse for not even evaluating it on an appropriate scale. I disagree that "that should be a necessary requirement to do research".
> > > >
> > > > Based on all the above reasons, I don't think the paper is ready (in the current form or the form that can be expected in the near future) to be accepted for a top-tier venue. I decide to keep my rating.
> > > >
> > > > [1] Michael D. Moffitt and Pratik Fegade. 2025. The ASPLOS 2025 / EuroSys 2025 Contest on Intra-Operator Parallelism for Distributed Deep Learning. In Proceedings of the 30th ACM International Conference on Architectural Support for Programming Languages and Operating Systems, Volume 3 (ASPLOS '25). Association for Computing Machinery, New York, NY, USA, 5–17. https://doi.org/10.1145/3676642.3736399
> > > >
> > > > [2] Zheng, Lianmin, Zhuohan Li, Hao Zhang, Yonghao Zhuang, Zhifeng Chen, Yanping Huang, Yida Wang et al. "Alpa: Automating inter-and {Intra-Operator} parallelism for distributed deep learning." In 16th USENIX Symposium on Operating Systems Design and Implementation (OSDI 22), pp. 559-578. 2022.
> > > >
> > > > [3] Lin, Zhiqi, Youshan Miao, Quanlu Zhang, Fan Yang, Yi Zhu, Cheng Li, Saeed Maleki et al. "{nnScaler}:{Constraint-Guided} Parallelization Plan Generation for Deep Learning Training." In 18th USENIX Symposium on Operating Systems Design and Implementation (OSDI 24), pp. 347-363. 2024.

---

### Official Review · Reviewer_oxsL · 2025-10-30

**Soundness:** 3
**Presentation:** 2
**Contribution:** 3
**Rating:** 6
**Confidence:** 4

**Summary:**

This paper proposes a formal framework for integrating soft constraints—constraints that can be selectively relaxed with associated penalties—into systems that require strong guarantees of correctness and safety. The method achieves promising results on several verification and optimization tasks, showing that it can flexibly handle trade-offs between strict guarantees and optimization goals.

**Strengths:**

1. This manuscript is well organized, presenting a logically coherent structure .
2. The paper tackles an important and challenging problem, and addresses an interesting topic .
3. The proposed framework is theoretically sound and shows strong empirical outcomes.

**Weaknesses:**

1. Lack of interpretability or analysis of mechanism. The paper reports strong results but does not explain why the method performs well. A deeper ablation and theoretical intuition would improve clarity.
2. Methodological nature unclear. The approach appears algorithmic rather than learning-based. It would help if the authors clarified whether any learning or adaptation is involved, or whether the framework is purely a deterministic optimization procedure.

**Questions:**

1. Could the authors provide a more detailed explanation of why your method achieves such good results?
2. Can the authors clarify how the system differs from, or could potentially connect to, deep learning–based constrained optimization methods?

---

> ### Author Response · Authors · 2025-11-21
>
> Many thanks for your review. We first want to note that our method is primarily designed to optimize the execution of distributed deep learning workloads. While we believe it has potential beyond that, our experiments only verify its performance on distributed deep learning workloads.
>
> 1. Could the authors provide a more detailed explanation of why your method achieves such good results?
>
> You have a good point we will include a detailed explanation in the updated version of the paper, that we will upload in the next days.
> The computational costs of distributed deep learning workloads, strongly depend on their computational graph. Optimizing the partition and communication costs without memory constraints naturally leads to cost function networks, which heavily exploit the graph structure of the optimization objective. Therefore, they achieve great performance on the unconstrained task. However, adding the necessary memory constraint introduces large sum constraints as thousands of nodes can be active at the same time and their combined memory usage must not exceed the available memory. These constraints quickly make this problem hard for discrete optimizers like ILP solvers (Gurobi, Cplex) or constrained based solvers like Google OR-Tools CP-SAT solver which is currently used in XLA for the same task. We sidestep this issue, by introducing penalty terms for every variable individually. If a sum constraint is violated, the penalty for all variables in the violated constraints are increased until the constraint is satisfied. This method exploits the excellent performance of cost function network optimizers on the unconstrained optimization task, while still efficiently ensures constraint satisfaction. You mention that you would like to see a deeper ablation, can you suggest some concrete results or further experiments that would be interesting to you?
>
> 2. Can the authors clarify how the system differs from, or could potentially connect to, deep learning–based constrained optimization methods?
>
> We solve a problem in distributed deep learning using classic optimization techniques. Of course, a deep learning based approach might be possible and could lead to further speedups. The biggest potential for a deep learning based system together with our approach would be in a better weight initialization and tuning.

---

### Official Review · Reviewer_hu4K · 2025-11-01

**Soundness:** 3
**Presentation:** 1
**Contribution:** 2
**Rating:** 2
**Confidence:** 5

**Summary:**

This paper presents a top-performing solution to the ASPLOS 2025 Contest on Intra-Operator Parallelism for Distributed Deep Learning. The authors propose a solver based on usage-constrained relaxation, where memory requirements are incorporated as soft constraints within the cost model rather than enforced as hard limits. The method uses adaptive weight tuning and greedy post-processing to efficiently generate feasible, low-cost solutions across large computational graphs. The approach achieves state-of-the-art results, significantly outperforming commercial compilers such as XLA and even approaching theoretical lower bounds on several benchmarks.

**Strengths:**

1. The idea of encoding memory requirements as soft constraints with adaptive penalties is both simple and elegant. It transforms a difficult combinatorial optimization problem into a tractable form while maintaining feasibility.
2. The paper provides extensive experimental validation, including comparisons with commercial compilers and exact solvers, thorough convergence and ablation studies, and a detailed analysis of contest benchmarks.

**Weaknesses:**

1. The paper reads more like a technical report rather than a research paper. The narrative emphasizes implementation details and experimental results but gives less attention to the high-level conceptual intuition behind the approach.
2. Much of the background context and some of the benchmark and contest results overlap with material already presented in the official ASPLOS contest report.
> Moffitt, Michael D. and Fegade, Pratik, "The ASPLOS 2025 / EuroSys 2025 Contest on Intra-Operator Parallelism for Distributed Deep Learning", Proceedings of the 30th ACM International Conference on Architectural Support for Programming Languages and Operating Systems, 2025.

**Questions:**

1. How many variables $x_i$ are there in each $c_i$?
2. Why does each node i have a separate $w_i$? It seems $w_i$ is global from the algorithm.
3. For the post-processing step, does the solver traverse the entire computation graph to identify candidate nodes, or is there a targeted strategy for selecting which nodes to revisit?
4. It would be informative to include a latency breakdown across different stages of the solver (e.g., preprocessing, relaxation solving, and greedy refinement) to better understand where most computation time is spent.
5. How sensitive is the final performance to the initial solution? Does the adaptive tuning always converge to similar-quality results regardless of the starting point?

---

> ### Author Response · Authors · 2025-11-21
>
> Thank you for your thoughtful review. We address your concerns and questions point by point below.
>
> **Focus of the paper and background**
> This is a good suggestion.
> We believe that it is necessary to include a full description of the formal problem in our paper,
> but we agree that the background section could be shortened and additional material moved to the appendix.
> We will use the space to highlight the main contribution and conceptual ideas and upload a new version in the next days. The large overlap with the report about the contest is natural and we got full permission to use the same graphics in our paper by their creators. We refrained from citing their paper in the anonymous version, because it would reveal our identities. Instead we just link to the GitHub repository for now, as it contains all relevant resources. The contribution of our paper is our method and its experimental analysis.
>
>
> **Questions**
> 1. How many variables $x_i$ are there in each $c_i$?
>
> Thank you for this question, this was probably not clear enough. $D_i$ is a set of execution strategies of node $i$, the whole cost function is defined on Cartesian product of the $D_i$ and given as a sum of univeriate functions $c_i$, that are defined on the corresponding $D_i$ and bivariate function $c_i,j$ that are defined on $D_i \times D_j$. We make this clearer in the updated paper.
>
> 2. Why does each node i have a separate  $w_i$
>
> There are time intervals with low and high memory usage. Nodes that are only in low memory usage intervals, can use more memory and thus need lower penalties. Because Algorithm 1, only increases the penalty for nodes in unfeasible intervals in Step 7, the penalties diverge from the global value set during initialization. The first data point for each instance in Figure 6 represents the first feasible solution with uniform weights. On more difficult instances, the non uniform weights allow to find up to 2x better solutions. We update the paper accordingly.
>
>
> 3. For the post-processing step, does the solver traverse the entire computation graph to identify candidate nodes, or is there a targeted strategy for selecting which nodes to revisit?
>
> We iterate over all nodes until improvement is no longer possible or a timeout of 1s is reached. Since the individual checks are very fast, this procedure usually takes less than a second. Even on the largest instance, Y, 1s is sufficient to improve the solution by 8% during the first iterations as shown in Figure 8 in the appendix.
>
> 4. It would be informative to include a latency breakdown across different stages of the solver (e.g., preprocessing, relaxation solving, and greedy refinement) to better understand where most computation time is spent.
>
> We have not included this information, because almost all time is spent on relaxation solving. However, since we are restructuring the paper, if space permits we will address this point as outlined in the following. We only do minimal preprocessing while reading the input file, thus the overhead of preprocessing is negligible. The greedy refinement usually takes less than a second, but is limited to one second as mentioned above. Thus, the vast majority of the time is spent solving the relaxed problem for different penalties.
>
> 5. How sensitive is the final performance to the initial solution? Does the adaptive tuning always converge to similar-quality results regardless of the starting point?
>
> This is a very good point and we address it in the updated paper. As shown in Figure 6, our algorithm is very robust across 10 random seeds. The 95% confidence intervals indicated by the shaded regions are narrow.

---

> > ### Comment · Reviewer_hu4K · 2025-11-28
> >
> > I appreciate the authors' efforts in clarifying my questions and revising the manuscript. However, the scope of the paper still feels limited to the contest setting. Although related systems such as Alpa and Ligra are cited, they are not included in the evaluation, and it remains unclear how the proposed approach compares end-to-end with state-of-the-art sharding systems in XLA, such as Alpa, or more recent works like PartIR [1] and Shardy [2].
> >
> > > "The large overlap with the report about the contest is natural and we got full permission to use the same graphics in our paper by their creators."
> >
> > Even with explicit permission from the contest organizers, I believe the authors should still consider reproducing the figures or using alternative examples to reduce the degree of similarity and mitigate concerns about dual submission (if authored by the same organizer) or plagiarism (if authored by different people). Given that the updated manuscript contains substantial new content, ICLR should not regard it as the "same submission". Nevertheless, I encourage the authors to communicate closely with the contest organizers and consider submitting the work to ASPLOS, where the contest context is more appropriate and where larger overlaps with the contest report would be better aligned with the venue's expectations.
> >
> > [1] Sami Alabed, Daniel Belov, Bart Chrzaszcz, Juliana Franco, Dominik Grewe, Dougal Maclaurin, James Molloy, Tom Natan, Tamara Norman, Xiaoyue Pan, Adam Paszke, Norman A. Rink, Michael Schaarschmidt, Timur Sitdikov, Agnieszka Swietlik, Dimitrios Vytiniotis, and Joel Wee. 2025. PartIR: Composing SPMD Partitioning Strategies for Machine Learning. In Proceedings of the 30th ACM International Conference on Architectural Support for Programming Languages and Operating Systems, Volume 1 (ASPLOS '25). Association for Computing Machinery, New York, NY, USA, 794–810. https://doi-org.proxy.library.cornell.edu/10.1145/3669940.3707284
> >
> > [2] OpenXLA, Shardy, https://github.com/openxla/shardy

---

> ### Author Response · Authors · 2025-11-28
>
> Thank you for continuing the discussion. Since Alpa has been integrated into XLA, we effectively do compare against Alpa on intra-operator parallelism. For the other methods we would need to do end-to-end evaluations, however our method only becomes interesting for large device meshes. The smallest mesh in the competition has 32 accelerators and the much more interesting meshes contain 128 accelerators. Unfortunately, we lack the resources to run end-to-end experiments on device meshes at that scale. Moreover, none of the other methods ever compares themselves to Alpa, so there is no indication of how they perform compared to each other.
>
> We will consider your further suggestions in another revision.

---

### Author Response · Authors · 2025-11-28
**Updated Paper**

We thank the reviewers again for their valuable feedback. We updated the paper to address the following points:

- We explained the problem we are solving in more detail and connect the theory better to the practical workloads.
- We focus on a more high level description and removed implementation details. This includes a section about the Lagrangian relaxation, why cost function network optimization is so effective in this setting and why we defer from the standard Lagrangian relaxation.
- We obtained sharper lower bounds for all instances, which proves that our solver finds near optimal distribution schedules that will speed up deep learning workloads.

Finally we want to highlight again that as written in the official contest report [1], the problem is highly relevant for anyone running distributed deep-learning workloads:

> A chief enabler of large-scale deep learning is the distribution of computation across multiple interconnected hardware
accelerators. In order to unlock the maximum possible performance, a compiler must first select a reasonable strategy
to parallelize a model’s operations.

And the problem of efficiently optimizing inter-operator parallelism was still open before the contest:
> Although a variety of techniques have been considered (including random sampling, constraint programming, Monte Carlo Tree Search, etc.), it remains an open question which is the most efficient and effective way to produce high-quality model partitionings at compile time.

In our paper we answer this open question for inter-operator parallelism as our solver quickly finds near-optimal solutions.



[1] Michael D. Moffitt and Pratik Fegade. 2025. The ASPLOS 2025 / EuroSys 2025 Contest on Intra-Operator Parallelism for Distributed Deep Learning. In Proceedings of the 30th ACM International Conference on Architectural Support for Programming Languages and Operating Systems, Volume 3 (ASPLOS '25). Association for Computing Machinery, New York, NY, USA, 5–17. https://doi.org/10.1145/3676642.3736399

---

### Meta-Review · Area_Chair_7Dj1 · 2026-01-05

**Summary:**

Strength: The problem in consideration is important and timely. The paper presents extensive experimental results.

Weakness: The paper presentation needs to be improved. The method needs a better explanation. Also, the experimental results need to be strengthened. The technical novelty also needs to be better highlighted.

**Reviewer Concerns:**

Weakness: The paper presentation needs to be improved. The method needs a better explanation. Also, the experimental results need to be strengthened. The technical novelty also needs to be better highlighted.

**Reviewer Scores:**

Based on the comments and discussions, the reviewers will likely keep their scores.

---

### Decision · Program_Chairs · 2026-01-26

Reject